# Perspectives of Quantitative GC-MS, LC-MS, and ICP-MS in the Clinical Medicine Science—The Role of Analytical Chemistry

**DOI:** 10.3390/jcm13237276

**Published:** 2024-11-29

**Authors:** Dimitrios Tsikas

**Affiliations:** Core Unit Proteomics, Institute of Toxicology, Hannover Medical School, 30623 Hannover, Germany; tsikas.dimitros@mh-hannover.de

**Keywords:** analytical chemistry, clinical medicine, gas chromatography, life sciences, liquid chromatography, mass spectrometry

## Abstract

Mass spectrometry (MS) is the only instrumental analytical technology that utilizes unique properties of matter, that is, its mass (*m*) and electrical charge (*z*). In the magnetic and/or electric fields of mass spectrometers, electrically charged native or chemically modified (millions) endogenous and (thousands) exogenous substances, the analytes, are separated according to their characteristic mass-to-charge ratio (*m*/*z*) values. Mass spectrometers coupled to gas chromatographs (GC) or liquid chromatographs (LC), the so-called hyphenated techniques, i.e., GC-MS and LC-MS, respectively, enable reliable determination of the concentration of analytes in complex biological samples such as plasma, serum, and urine. A particular technology is represented by inductively coupled plasma-mass spectrometry (ICP-MS), which is mainly used for the analysis of metal ions. The highest analytical accuracy is reached by using mass spectrometers with high mass resolution (HR) or by tandem mass spectrometers, as it can be realized with quadrupole-type instruments, such as GC-MS/MS and LC-MS/MS, in combination with stable-isotope labeled analytes that serve as internal standards, like a standard weight in scales. GC-MS belongs to the oldest and most advanced instrumental analytical technology. From the very beginning, GC-MS found broad application in basic and applied research sciences. GC-MS has played important roles in discovering biochemical pathways, exploring underlying mechanisms of disease, and establishing new evidence-based pharmacological therapy. In this article, we make an inventory of the use of instrumental mass spectrometry in the life sciences and attempt to provide a perspective study on the future of analytical mass spectrometry in clinical science, mainly focusing on GC-MS and LC-MS. We used information freely available in the scientific database PubMed (retrieved in August–November 2024). Specific search terms such as GC-MS (103,000 articles), LC-MS (113,000 articles), and ICP-MS (14,000 articles) were used in the Title/Abstract in the “PubMed Advanced Search Builder” including filters such as search period (1970–2024). In total, around 103,000 articles on GC-MS, 113,000 articles on LC-MS (113,000), and 14,000 articles on ICP-MS were found. In the period 1995–2023, the yearly publication rate accounted for 3042 for GC-MS articles and 3908 for LC-MS articles (LC-MS/GC-MS ratio, 1.3:1). Our study reveals that GC-MS/MS, LC-MS/MS, and their high-resolution variants are indispensable instrumentations in clinical science including clinical pharmacology, internal and forensic medicine, and doping control. Long-tradition manufacturers of analytical instruments continue to provide increasingly customer-friendly GC-MS and LC-MS apparatus, enabling fulfillment of current requirements and needs in the life sciences. Quantitative GC-MS and GC-MS/MS methods are expected to be used worldwide hand in hand with LC-MS/MS, with ICP-MS closing the gap left for metal ions. The significance of analytical chemistry in clinical science in academia and industry is essential.

## 1. Introduction

Mass spectrometry coupled to gas chromatography (GC-MS) and liquid chromatography (LC-MS) are widely used instrumental techniques for the reliable quantitative determination of numerous endogenous and exogenous substances, the analytes, in biological samples such as plasma, urine, waters, and air, in modern life sciences. Quantitative determination means measuring the concentration of the analytes in a given biological sample. Knowledge of the concentration of analytes is of particular importance in health and disease, pharmacotherapy, supplementation, sports, and lifestyle [1,2,3,4,5,6,7,8,9,10,11,12,13].

Modern GC-MS, GC-MS/MS, LC-MS, and LC-MS/MS apparatus are computer-operated and controlled [14,15,16,17]. A schematic presentation of GC-MS and GC-MS/MS instruments based on the quadrupole technology is illustrated in Figure 1. GC-MS and GC-MS/MS instruments consist of an autosampler, a gas chromatograph (GC), a mass spectrometer (MS), a detector, pumps, and a computer, which controls all components. The GC and MS instruments are connected via the interface, which is kept at a high temperature (e.g., 300 °C). In LC-MS and LC-MS/MS apparatus, a liquid chromatograph (LC) is used instead of a gas chromatograph, and electrospray ionization (ESI) instead of electron ionization (EI) or chemical ionization (CI).

As a rule, quantitative analysis is a multi-step process, even by using analytical mass spectrometry, which possesses inherent accuracy and, therefore, high analytical reliability in terms of accuracy and precision. The basic principle of instrumental mass spectrometry, such as GC-MS and GC-MS/MS (Figure 1) as well as LC-MS and LC-MS/MS, is their separation capability of electrically charged inorganic (e.g., nitrate) and organic chemical substances (e.g., glucose, amino acids, proteins) in electric or magnetic fields according to their mass-to-charge (*m*/*z*) ratio (Figure 1). Due to this unique feature, mass spectrometry is the sole analytical technique that can utilize analytes labeled with heavy stable isotopes (mostly ^2^H, ^13^C, ^15^N), the isotopologs, as internal standards, kind of standard weights as used in scales (balances). Analyte and its internal standard have almost identical physicochemical properties, except for their different *m/z* values, e.g., *m*/*z* 46 for ^14^N-nitrite and *m*/*z* 47 for ^15^N-nitrite, which are separated by mass spectrometry [18,19,20,21,22].

The aim of the present article is to make an inventory of the use of instrumental mass spectrometry in the life sciences from the past decades until the present day and to attempt to provide a perspective study on the near future of analytical mass spectrometry in clinical science. Information freely available in the scientific database PubMed was used. Our focus was on the use of quantitative methods based on GC-MS, GC-MS/MS, LC-MS, LC-MS/MS, and their high-resolution variants, GC-HRMS and LC-HRMS. Inductively coupled plasma-mass spectrometry (ICP-MS) is a particular and unique MS-based technology that allows the analysis of metal ions, an analytical gap left by GC-MS and LC-MS [23].

## 2. Methods

PubMed^®^ comprises more than 37 million citations for biomedical literature from MEDLINE, life science journals, and online books. Citations may include links to full-text content from PubMed Central and publisher websites (https://pubmed.ncbi.nlm.nih.gov). The PubMed data bank can be searched in many ways (Figure 2). In the “PubMed Advanced Search Builder”, search terms can be combined, e.g., GC-MS with NEONATES, that can be searched within all parts of articles, or specifically in selected articles parts and “All Fields” such as in Title and Abstract (Title/Abstract), as shown below, also setting filters such a search period. The software provides a list of found articles and a Figure, the data of which can be exported (.csv) and further analyzed when desired. In the present work, exported data were analyzed by GraphPad Prism 7.0 (GraphPad Software, San Diego, CA, USA) (see Figure 3). Other inclusion and exclusion criteria were not used. It should be emphasized that such searches are expected to be “rough”.

## 3. Results and Discussion

### 3.1. Evolution of Mass Spectrometry in Science

Articles published in PubMed were used to estimate potential trends of the use of GC-MS in comparison to the use of the orthogonal LC-MS (Figure 3). In the period 1995–2023, the publication rate of GC-MS articles in PubMed was almost linear, with an estimated yearly rate of 3042. In the closely comparable period 1997–2023, the publication rate of LC-MS articles was also linear, with a higher estimated yearly rate of 3908 (LC-MS/GC-MS ratio, 1.3:1). There seems not to be a “saturation” effect.

After a considerable gap over several years, the use of LC-MS-MS increased remarkably and reached the rate by which GC-MS is used so that the difference between the two methodologies is practically zero over several decades. However, it should be noted that only about 5% of the GC-MS articles applied GC-MS-MS (Σ 2725), in contrast to at least 60% of the LC-MS articles actually applied LC-MS-MS. In the period 2000–2024, there was a linear increase in the number of PubMed articles for LC-MS-MS (*r*^2^ = 0.9252) and GC-MS (*r*^2^ = 0.9187), with slope values corresponding to about 3400 and 3000 articles per year, respectively. In the first seven months of the year 2024, around 4000 GC-MS-related articles and 6000 LC-MS-related articles appeared in PubMed (LC-MS/GC-MS ratio, 1.5:1).

### 3.2. Utilization of Mass Spectrometry All over the World

Table 1 shows that the GC-MS-, LC-MS-, and ICP-MS-based technologies are used worldwide. ICP-MS is generally less frequent, presumably because of its high selectivity for metal ions. Yet, this analysis is not standardized, e.g., for the number of residents and for other economic factors of the countries. The instrumentation of laboratories is primarily a question of money. Quantitative GC-MS and GC-MS-MS (cheaper) are expected to be used worldwide hand in hand with LC-MS-MS (more expensive), presumably with preference for some countries.

### 3.3. Utilization of Mass Spectrometry Across Human Life

We focused on GC-MS and GC-MS/MS and searched PubMed using the term “gas chromatography-mass spectrometry humans” in combination with a variable term, including diseases, syndromes, clinical trials, and other conditions. The second variable search term was selected based on analyses performed by GC-MS and GC-MS/MS in our laboratory in the framework of scientific cooperation of the host institute of Clinical Pharmacology and later of the Institute of Toxicology of the Hannover Medical with departments of host and foreign clinical universities. Core research topics included cardiovascular and renal diseases of adults and children (pediatrics). The results are listed in Table 2.

Historically, the host laboratory was gradually specialized in the quantitative GC-MS and GC-MS/MS analysis of eicosanoids (e.g., prostaglandins, thromboxane, leukotrienes, endocannabinoids), members of the L-arginine/NO pathway (e.g., nitrite and nitrate), post-translational modifications (e.g., ADMA, DMA), biogenic amines and polyamines (e.g., histamine, spermidine), oxidative stress (notably, MDA and 8-*iso*-PGF_2α_), and oxidized and nitrated fatty acids including oleic acid. The analytical spectrum of the host laboratory also included quantitative GC-MS and GC-MS/MS analysis of certain drugs, including paracetamol (acetaminophen), acetylsalicylic acid (aspirin), and ibuprofen. Due to the indispensable utility of urinary creatinine for correction of analyte concentrations measured in urine samples collected by spontaneous micturition, GC-MS and GC-MS/MS methods for the measurement of urinary and circulating creatinine were developed, validated, and used in clinical studies. For the sake of simplicity, creatinine has not been considered in Table 2. Obviously, GC-MS and GC-MS/MS have been widely used worldwide in various diseases and conditions, such as diabetes and COVID-19, in pediatrics, occupational medicine, and nutrition.

Table 3 suggests that GC-MS- and LC-MS-based methodologies have been applied in basic and applied research for the quantitative determination of numerous classes of analytes virtually in all areas of modern human life, including medicine, pharmacology, toxicology, pharmacy, the food industry, the environment, epidemiology, chemistry, and biochemistry.

### 3.4. Gas Chromatography-Mass Spectrometry and Liquid Chromatography-Mass Spectrometry–Foe or Friends?

Studies reporting on the comparison of GC-MS with GC-MS/MS methods are very rare and limited to very few analyte classes. A recent direct comparison between GC-MS and GC-MS/MS analysis of disinfection byproducts in drinking water revealed that the limit of quantitation (LOQ) values achieved with GC-MS/MS were up to 50 times lower than those obtained by GC-MS [24].

Studies reporting on the comparison of GC-MS- with LC-MS-based methods are even more rare and limited to very few analytes and researcher groups. The lack of comparability between GC-MS/MS and LC-MS/MS methods for individual analytes or groups of analytes, such as for many eicosanoids, notably prostaglandins, thromboxane, and their metabolites, could be a sign of lacking reliability of LC-MS/MS methods for those analytes [25,26]. For example, the concentration of thromboxane B_2_ (TxB_2_) has been reported to be about 3 pg/mL plasma by GC-MS compared to 179 pg/mL by LC-MS/MS, and the concentration of its major metabolite 11-dihydro-TxB_2_ 1–2 pg(mL by GC-MS, 0.8–2.5 pg/mL by GC-MS/MS, but 179 pg/mL by LC-MS/MS (see Refs. 15, 17, 37, 40 in Ref. [25]).

Cross-validation of thoroughly validated LC-MS/MS methods with thoroughly validated GC-MS/MS or GC-MS methods analytes is the most useful approach to check and ensure analytical certainty but is also rarely performed [27]. As an example, previously validated LC-MS/MS [28] and GC-MS/MS [29] methods for the endocannabinoid anandamide (AEA) have been cross-validated by our group. Figure 4 shows that both linear regression analysis and the Bland-Altman procedure show remarkable agreement between the LC-MS/MS method and the GC-MS/MS method for AEA in human plasma. LC-MS/MS provided, on average, 16% lower values than GC-MS/MS. This order of discrepancy is low considering that cross-validation has not been originally planned, and LC-MS/MS analyses were performed one year later than the GC-MS/MS analyses. It is notable that the concentration of AEA in the plasma of healthy humans is of the order of 1 nM [28,29].

### 3.5. Examples of GC-MS Applications in Clinical Science

Clinical trials are prospective biomedical or behavioral research studies on human participants designed to answer specific questions about biomedical or behavioral interventions, including new treatments such as novel vaccines and drugs and known interventions that warrant further study and comparison. Clinical trials generate data on dosage, safety, and efficacy (https://www.nih.gov/health-information/nih-clinical-research-trials-you/basics). Clinical studies are conducted after approval by ethics committees. They are accompanied by different kinds of clinical and biochemical analyses. Below, two examples of clinical trials in adults and children are described in which biochemical analyses of particular endogenous and exogenous substances were performed by quantitative GC-MS and GC-MS/MS in plasma, serum, and urine samples of the study participants [30,31].

#### 3.5.1. Patients with Becker Muscular Dystrophy

Becker muscular dystrophy (BMD) is an X-linked recessive disorder that affects one in 18,000 male births and is a milder form of Duchenne muscular dystrophy (DMD). A single-center, open-label, proof-of-concept study, approved by the local Ethics Committee and National Swiss Drug Agency, was conducted following all laws and guidelines of the Humanforschungsgesetz of Switzerland and the ICH-GCP rules. Twenty ambulatory BMD patients were recruited from the neurologic and neuro-paediatric outpatient clinics in Basel and from the German and Swiss patient registries (www.treat-nmd.de/register and www.muskelkrank.ch). Inclusion criteria were as follows: genetically or immunohistochemically confirmed diagnosis, ambulant at inclusion without walking aids, and older than 18 years of age at inclusion. Exclusion criteria were intake of supplementary L-arginine, L-citrulline, tetrahydrobiopterin, or metformin within the last three months, other significant concomitant illness with impairment of renal, hepatic, respiratory, or cardiac function or malignancy, known hypersensitivity to study medication and participation in any therapeutic trial within three months prior to inclusion. Patients who met all inclusion criteria and did not meet any of the exclusion criteria were subsequently allocated in a 1:1 ratio to receive metformin (Met, MET group) or L-citrulline (Cit, CITR group) as a first treatment, followed by the combination of the two investigational products. All patients in the study were male and did not differ with respect to their age.

The study design is illustrated in Figure 5. All analytes in the study were determined by previously reported GC-MS methods. The analyte concentrations measured in the serum samples from this study are summarized in Table 4.

The main outcome of this study, from a biochemical perspective, is that metformin or L-citrulline supplementation to BMD patients resulted in remarkable antidromic changes of two biochemical pathways in which hArg and GAA are biosynthesized. In combination, metformin and L-citrulline at the doses used in the present study seem to abolish the biochemical effects of the single drugs in slight favor of L-citrulline. In all serum samples of the participants who took metformin, this drug was identified and quantitated by GC-MS [30].

#### 3.5.2. The Arginine Test

Children with short stature but not with other endocrinologic, organic, chromosomal, metabolic, or psychosocial reasons for macrosomia were routinely examined for growth hormone deficiency (GHD) by the so-called L-arginine test. The arginine test in pediatrics includes a 30-min lasting infusion of a concentrated L-arginine (Arg) solution in physiological saline (i.e., 0.5 g Arg per kg bodyweight).

Figure 6 shows an example of the application of GC-MS and GC-MS/MS methods in a clinical pediatrics trial from a biochemical perspective [32]. The arginine test resulted in high circulating free Arg concentrations and allowed studying acute effects of Arg and its metabolites that may be rapidly formed from Arg, including nitric oxide (NO), asymmetric dimethylarginine, and homoarginine (Figure 4) [33].

### 3.6. ICP-MS in Clinical Science

ICP-MS is the method of choice for the analysis of elements in complex biological systems, including biomonitoring of major, trace, and toxicologically relevant elements (reviewed in Ref. [23]). Its online coupling to laser ablation and chromatography expanded the scope and application range of ICP-MS to quantitative speciation analysis and element bioimaging. ICP-MS emerged as a versatile technique with a vast potential to provide complementary perspectives in medical disciplines [34,35,36,37,38,39,40]. About 1% to 10% of the articles found in PubMed (Table 1) refer to the application of ICP-MS in clinical studies all over the world.

## 4. Conclusions and Perspectives

### 4.1. The Place of Mass Spectrometry in Clinical Science

Good science needs good mass spectrometry [41]. Analytical mass spectrometry techniques such as GC-MS and LC-MS require deep knowledge of analytical chemistry, of known and potential pitfalls in particular analytical processes, notably in the quantitative analysis of endogenous analytes, and especially of those that occur at extremely low concentrations such as many eicosanoids.

As a rule, chemist analysts are performing mass spectrometry analyses, the majority of whom presumably have not graduated in academic Analytical Chemistry. Today’s Analytical Chemistry is not only chemistry [42]. Chemical analysis experience made with some analytes is not always easily transferable to other analytes, even if they are structurally closely related, without previous examination and testing. Chemist analysts and chemists who are working in chemical analysis laboratories need to be comprehensively informed about already available methods based on GC-MS (and LC-MS) for their particular analytes in order to be able to find out the most suitable procedures. This information is available on the internet. With respect to GC-MS, the use of GC-HRMS and GC-MS/MS should be preferred where feasible in quantitative analyses of biological samples.

By nature, the spectrum of analytes that are accessible to LC-MS/MS is much wider than that to GC-MS and GC-MS/MS. With few exceptions, there are no general rules regarding the selection of analytes/classes of analytes that can be analyzed exclusively by LC-MS/MS or GC-MS(/MS). The vast majority of analytes require derivatization for GC-MS-based analysis. Thus, even high polar and hydrophilic substances such as glucose 6-phosphate can be analyzed by GC-MS after suitable chemical derivatization [43]. On the other hand, phase II metabolites such as glucuronides and sulfates can be analyzed by GC-MS after enzymatic removal of the glucuronide and sulfate residues, respectively. There are also many examples of the equally good utility of GC-MS and LC-MS methods for the same analytes, such as cytosine metabolites in samples from clinical trials [44,45].

GC-MS and LC-MS are orthogonal analytical techniques due to their principal differences in chromatography, i.e., GC versus LC. In some respects, GC-MS and LC-MS may be antagonistic. Yet, analytical laboratories would be happy to have both capabilities in their portfolio. In the clinical laboratory, tandem mass spectrometry (MS/MS) is considered the Gold Standard [46].

A rather practical difference between GC-MS and LC-MS refers to automation. In LC-MS/MS, almost complete automation is possible, for instance, in Clinical Chemistry. The degree of automation in GC-MS is limited. Despite considerable technical improvements in automation, including sample extraction and derivatization techniques, quantitative GC-MS, GC-MS/MS, and GC-HRMS methods will remain, at best, semi-automated.

Irrespective of the analytical technique used, crucial concerns refer to a series of pre-analytical factors that include sample collection, storage until sample workup, and study design, even in placebo-controlled clinical studies. Non-consideration of pre-analytical issues is likely to distort the in vivo reality because of the generation of questionable analytical results [47].

### 4.2. Quantitative Analysis, Metabolomics, Proteomics and Regulatory Issues

By nature, LC-MS, but not GC-MS, is suitable for proteomic analyses, in which native peptides and proteins are analyzed qualitatively and, to some degree, quantitatively.

In contrast, GC-MS and LC-MS are equally useful in the so-called untargeted (i.e., qualitatively) metabolomics, i.e., analysis of many low-molecular-mass metabolites and their precursors from different biochemical pathways. Quantitative, i.e., the so-called targeted analytical methods based on GC-MS and LC-MS, have been reported for individual endogenous and exogenous analytes in plasma, urine, and other biological samples of healthy and diseased humans [48,49,50,51,52,53,54,55,56,57,58].

Targeted and untargeted metabolomics are associated with problems that have been recognized, and recommendations and protocols have been proposed for LC-MS/MS users [59]. Analogous steps are required in the use of “targeted GC-MS”, which is often synonymously used with quantitative GC-MS.

PubMed reveals comparable percentage values for GC-MS (22%) and 29% for LC-MS (29%) when searching for “FDA”, i.e., Food and Drug Administration. Regulatory and related issues such as method validation and method comparison refer mainly to drugs and foods, i.e., exogenous compounds. So, there is a gap in regulatory issues with respect to endogenous analytes, presumably the largest fraction of analytes.

Although lack of adherence to recommended (i.e., non-binding) nomenclature and guidelines in several areas of analytical chemistry is expected to persist further [21,60], official recommendations and guidelines are warranted, analogous to definitions in mass spectrometry [22].

A rather new phenomenon refers to the performance of commissioned analyses by untargeted and targeted analyses using GC-MS (GC-MS/MS, GC-HRMS) or LC-MS/MS in biological samples from clinical studies by commercial companies. This kind of service seems to be non-regulated.

The present survey examined the role of analytical mass spectrometry with the focus set on GC-MS and LC-MS. Historically, being the older and more mature instrumental analytical methodology, GC-MS, GC-MS/MS, and GC-HRMS have played essential roles in basic and clinical research in all areas of human life. GC-MS and LC-MS are complementary analytical technologies, with particular superiority on distinct topics. The results of the present survey suggest that quantitative GC-MS, GC-MS/MS, and LC-MS/MS techniques will be used worldwide in the near future in all areas of modern human life: medicine, pharmacology, toxicology, pharmacy, food industry, environment, and epidemiology, from several scientific perspectives including chemistry and biochemistry. There is a need for a more engaged role of regulatory authorities, organizations, and agencies, such as the World Anti-Doping Agency (WADA), in quantitative analysis of endogenous substances in biological samples collected in clinical studies, notably in metabolomics. It is expected that the significance of academic analytical chemistry and of chemist analysts in life science will further increase.

Eventually, as shown in the present work, GC-MS-, LC-MS- and ICP-MS-based analytical methods are widely used in the life sciences. As the primary aim of the survey was to discuss the importance of these techniques in clinical medicine sciences, the contribution of many researchers to other areas of research, such as in vitro studies and animal studies [61,62,63], could not be considered. Quantitative GC-MS and GC-MS/MS methods are expected to be used worldwide, hand in hand with LC-MS/MS. ICP-MS is unrivaled as it closes the gap left by the former for metal ions.

## Figures and Tables

**Figure 1 jcm-13-07276-f001:**
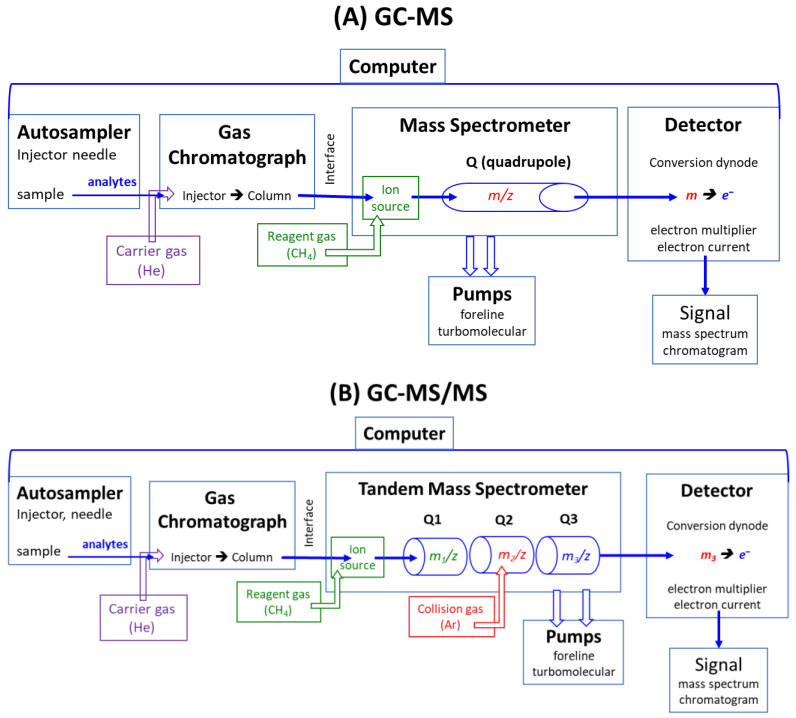
Simplified schematic presentation of modern quadrupole (Q)-based (**A**) gas chromatograph-mass spectrometers (GC-MS) and (**B**) gas chromatograph-tandem mass spectrometers (GC-MS/MS). In LC-MS and LC-MS/MS, a liquid chromatograph (LC) is used instead of a gas chromatograph (GC), and electrospray ionization is applied instead of chemical ionization or electron ionization (no use of reagent gases).

**Figure 2 jcm-13-07276-f002:**
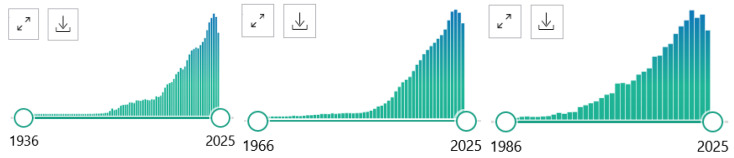
Original graphs showing the time course of publications found in PubMed by using the search terms “GC-MS” (left panel, 102,734 in total), “LC-MS” (middle panel, 113,171 in total), “ICP-MS” (right panel, 13,628). Retrieved on 13 November 2024.

**Figure 3 jcm-13-07276-f003:**
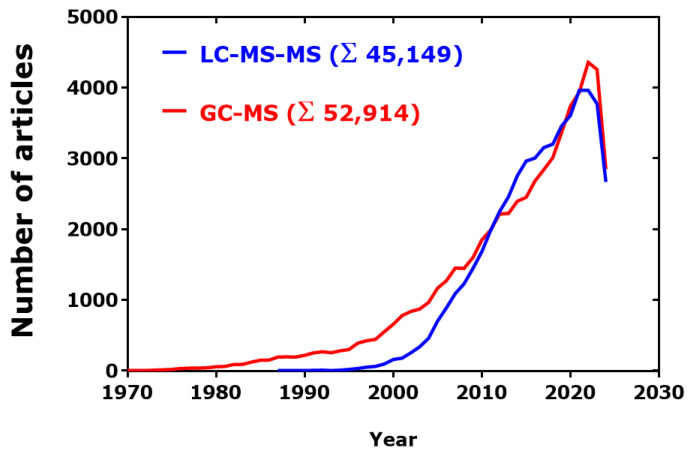
Time-course of publications found in PubMed by using the search terms “GC-MS[Title/Abstract]” and “LC-MS-MS[Title/Abstract].” Data were exported from PubMed and analyzed by GraphPad Prism 7.0 (GraphPad Software, San Diego, CA, USA). Retrieved on 17 August 2024.

**Figure 4 jcm-13-07276-f004:**
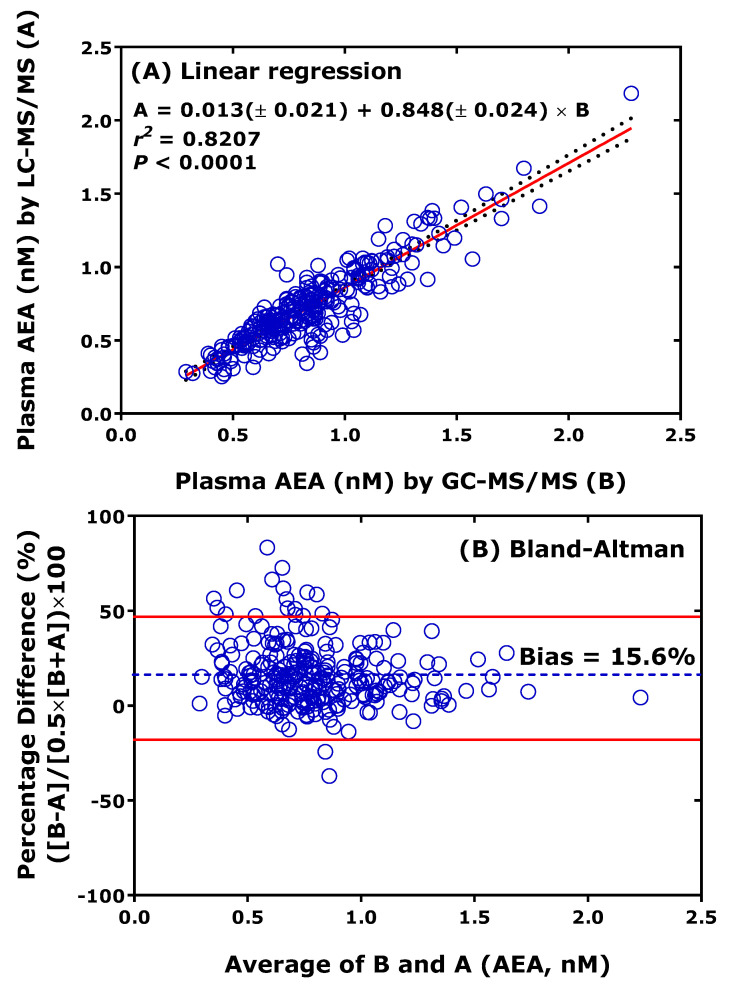
Comparison of measurements of anandamide (AEA) in 277 human plasma samples from a previously reported clinical trial performed by GC-MS/MS (method B) and LC-MS/MS (method A). LC-MS/MS analyses were performed about one year after the GC-MS/MS analyses. (**A**) Linear regression analysis and (**B**) Bland-Altman approach. Samples were analyzed for AEA on the instrument model TSQ 7000 by GC-MS/MS [29] and on the instrument model Xevo LC-MS/MS [28].

**Figure 5 jcm-13-07276-f005:**
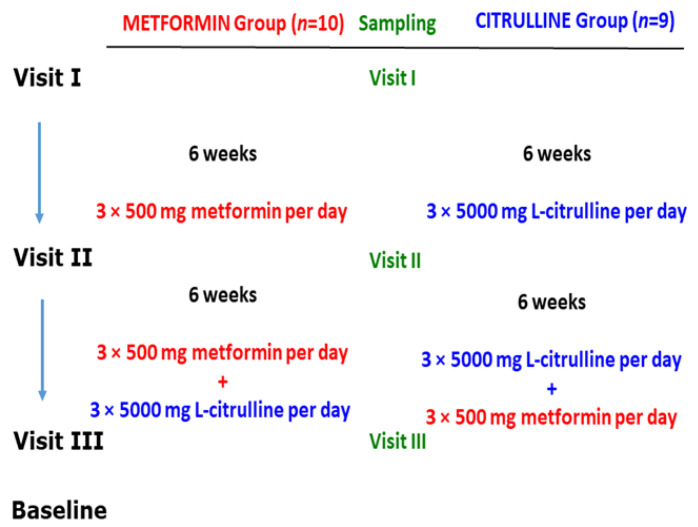
Study design on the effects of metformin and L-citrulline (for six weeks) and their combination (for six weeks) in patients with Becker muscular dystrophy (BMD). For more details, see the text. From Baskal et al., 2022 [30]; see also Hanff et al., 2018 [31].

**Figure 6 jcm-13-07276-f006:**
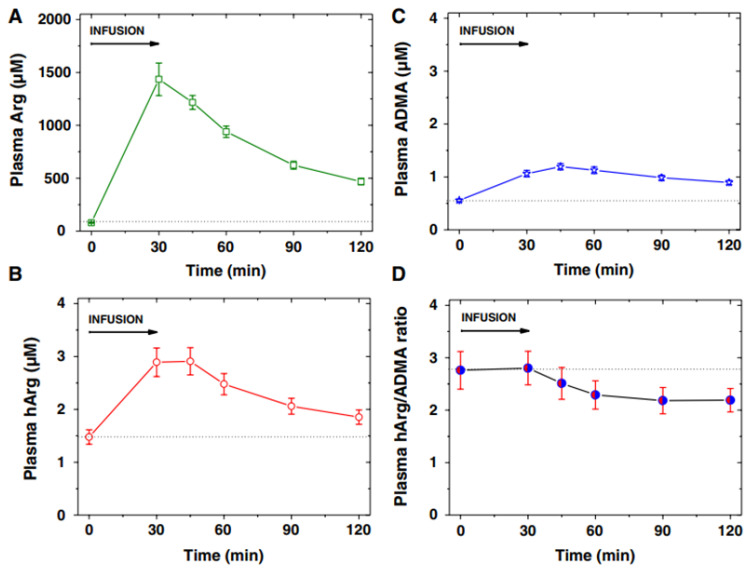
Plasma concentration (mean ± SEM) of (**A**) arginine (Arg), (**B**) homoarginine (hArg) and (**C**) asymmetric dimethylarginine (ADMA), and (**D**) plasma hArg/ADMA molar ratio at the start (0 min) and the end (30 min) of the L-arginine infusion (0.5 g/kg body weight) into 11 children, and at the indicated time points thereafter. Arg, hArg, and ADMA were measured by GC-MS-based methods. From Ref. [32].

**Table 1 jcm-13-07276-t001:** Number of publications in PubMed obtained by using the search terms “GC-MS”, “GC-MS-MS”, “LC-MS”, and “LC-MS-MS” (combined with GC-MS and LC-MS, respectively), and “ICP-MS” in combination with search term [continent] or [country] as indicated in alphabetical order. Retrieved on 17 August 2024.

Continent, Country	GC-MS	LC-MS	ICP-MS
AFRICA	1855	1305	286
AUSTRALIA	2110	2632	382
BRAZIL	2793	2597	488
CANADA	3210	4002	509
CHINA	16,863	23,018	2886
BELGIUM	1369	1879	254
FRANCE	3650	3870	707
GERMANY	6662	8016	1099
GREECE	930	897	96
ITALY	5371	4819	688
THE NETHERLANDS	1680	2974	136
SPAIN	4674	4596	956
UNITED KINGDOM	1674	2571	295
INDIA	4473	4436	362
JAPAN	5165	6251	715
REPUBLIC OF KOREA	1741	2266	212
RUSSIA	545	591	190
SINGAPORE	448	767	51
TURKIYE	1325	1269	308
USA	12,198	18,822	1626

**Table 2 jcm-13-07276-t002:** Number of articles found in PubMed by using the search term “gas chromatography-mass spectrometry humans” and the indicated search terms in alphabetical order. Retrieved on July 2024.

SEARCH TERM in PubMed	No. Articles
gas chromatography mass spectrometry	89,871
gas chromatography mass spectrometry humans	32,556
gas chromatography mass spectrometry humans ***AND***	
ATTENTION-DEFICITE HYPERACTIVITY DISORDER (ADHD)	24
ALLERGIC ASTHMA	27
ACUTE RESPIRATORY DISTRESS SYNDROME (ARDS)	43
ATOPIC DERMATITIS	34
ATOPIC DISEASE	21
BECKER MUSCULAR DYSTROPHY	6
CORONARY ARTERY DISEASE	77
COVID-19	146
CYSTIC FIBROSIS	76
DIABETES	953
DUCHENNE MUSCULAR DYSTROPHY	6
END-STAGE KIDNEY DISEASE	88
END-STAGE LIVER DISEASE	12
ERECTILE DYSFUNCTION	19
EXERCISE PHYSICAL	89
FETAL GROWTH RESTRICTION	40
GROWTH HORMONE DEFICIENCY	22
HELICOBACTER PYLORI	86
HYPERCHOLESTEROLEMIA	113
HYPOGONADISM	21
LACTATION	214
LIVER DISEASE	678
NEONATES	1298
NUTRITION	2185
OCCUPATIONAL MEDICINE	522
PERIPHERAL ARTERY OCCLUSIVE DISEASE	9
PHENYLKETONURIA	56
PREECLAMPSIA	46
PSORIASIS	57
RHEUMATIC DISEASE	92
ZELLWEGER SYNDROME	62

**Table 3 jcm-13-07276-t003:** Number of publications in PubMed obtained by using the search term “LC-MS” or “GC-MS” alone and in combination with the listed terms (in alphabetical order) in the indicated periods and in the years 2024 and 2025. Retrieved on 17 August 2024.

Search Term	Period	Total	2024 + 2025
LC-MS	1970–2025	111,412	6270
GC-MS	1970–2025	101,580	3967
*GC-MS AND* Amino acids	1968–2025	7123	216
Anti-doping	1987–2025	279	9
Biomarkers	1972–2024	5571	241
Drugs	1970–2025	14,038	595
Eicosanoids	1969–2024	2138	9
Fatty acids	1966–2025	13,288	468
Food	1968–2024	23,526	1345
Forensic	1971–2024	4817	138
Herbicides	1971–2024	1202	28
Hormones	1968–2024	5959	127
Lipids	1966–2024	25,378	818
Nutrition	1975–2024	5542	381
Occupational	1975–2025	1806	57
Pesticides	1970–2024	8068	267
Postmortem	1971–2024	1268	28
Supplements	1973–2024	1809	93
Vitamins	1968–2024	2225	60

**Table 4 jcm-13-07276-t004:** Serum concentrations (µM) and molar ratios of the biochemical parameters in the metformin (MET) and citruline (CITR) groups at baseline, after metformin (Met) or L-citrulline (Cit) treatment, and after combined treatment (Met + Cit). The Table was constructed with some data reported elsewhere [31].

	MET Group (*n* = 10)			CITR Group (*n* = 9)		
Analyte	Baseline	Met	Met + Cit	Baseline	Cit	Cit + Met
hArg	1.68 ± 0.60	1.31 ± 0.48 *	2.07 ± 0.45 ***	1.61 ± 0.35	2.70 ± 0.76 **	2.46 ± 1.00
GAA	2.00 ± 0.37	1.54 ± 0.46 **	2.55 ± 1.38 *	1.94 ± 0.26	2.87 ± 0.86 *	2.80 ± 1.60
Arg	129 ± 35.1	100.4 ± 22.8 *	174 ± 82.6 *	120 ± 19.5	220 ± 57.4 **	208 ± 86.3
ADMA	0.526 ± 0.142	0.496 ± 0.066	0.512 ± 0.095	0.528 ± 0.049	0.594 ± 0.087 *	0.600 ± 0.109
Nitrate	55.2 ± 17.1	50.0 ± 8.04	50.4 ± 9.52	56.6 ± 13.3	56.2 ± 20.1	60.3 ± 21.04
Creatinine	62.1 ± 18.0	58.3 ± 13.5	59.0 ± 13.9	49.8 ± 11.0	59.1 ± 20.5	52.2 ± 12.5
MDA	0.989 ± 0.350	0.780 ± 0.190 *	0.936 ± 0.370	0.884 ± 0.189	1.173 ± 0.404	1.081 ± 0.636
GAA/hArg	1.34 ± 0.51	1.34 ± 0.68	1.23 ± 0.56	1.26 ± 0.32	1.13 ± 0.46	1.17 ± 0.53
hArg/ADMA	3.19 ± 1.42	2.68 ± 1.07	4.15 ± 1.21 ***	3.07 ± 0.66	4.52 ± 0.92 **	4.07 ± 1.31
Nitrate/nitrite	20.6 ± 7.22	20.2 ± 5.15	20.7 ± 3.73	23.9 ± 5.10	22.1 ± 8.92	24.9 ± 8.6

Abbreviations: hArg, homoarginine; GAA, guanidinoacetate; ADMA, asymmetric dimethylarginine; MDA, malondialdehyde. Data are presented as mean  ±  SD. Statistical analysis was performed between baseline and after the first treatment values and between first treatment parameters and second treatment parameters using paired *t*-test and Wilcoxon rank test when appropriate. Asterisks indicate statistical difference (* *p* < 0.05; ** *p* < 0.01; *** *p* < 0.001).

## Data Availability

Data are available on request.

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
