# Peer review of "Perspectives of Quantitative GC-MS, LC-MS, and ICP-MS in the Clinical Medicine Science—The Role of Analytical Chemistry"

_jcm, 2024, doi:10.3390/jcm13237276_

Round 1
Reviewer 1 Report
Comments and Suggestions for Authors
Dear Author,
the manuscript entitled "Perspectives of Spectrometry in the Clinical Medicine Science - The Role of Analytical Chemistry" is interesting and well written. However, I have some questions that need to be addressed.
General comments
Introduction: this article is reported to be on spectrometry, but actually the manuscript only goes through hyphenated techniques.
The "Methods" section is missing. How did you perform the literature search reported as figure captions in the "Results and discussion" section? What search terms did you use? Did you use any exclusion/inclusion criteria?
The "Results and discussion" section is only about LC-MS and GC-MS techniques. However, there are many applications of mass spectrometry in analytical chemistry, such as the characterization of new compounds, the application of high-resolution mass spectrometry for metabolism studies (and the identification of unknown metabolites), the use of mass spectra libraries for compounds matching in screening analyses or the direct analysis in real time (DART).
Considering that the manuscript is supposed to be on the role of mass spectrometry in the analytical chemistry, too many information are missing.
Author Response
See PDF
Reviewer 2 Report
Comments and Suggestions for Authors
In the perspective manuscript, the author detailed the functional of mass spectrometry in clinical medicine science. The data of the perspective manuscript may enhance the understanding the research status of GC-MS, GC-MS/MS, LC-MS, and LC-MS/MS in clinical medicine application. Besides, the insight of the manuscript is outstanding with enough workload in mass spectrometry-associated investigation and summary. Therefore, the perspective manuscript is recommended to be accepted by the Journal of Clinical Medicine. Here are some concerns for the author:
1) For the title, the title is recommended to be revised as “Perspectives of Mass Spectrometry in the Clinical Medicine Science - The Role of Analytical Chemistry”.
2) In the 1st sentence of the 1st paragraph of the Introduction section, the author presented a wonderful description of application of MS. However, references are in need to support this sentence. One research in profiling exogenous medicines’ tissue distribution and excretion characteristics in life matrix (10.3389/fphar.2022.827668) and another study in endogenous 5-hmC in human urine (10.1021/ac5038895) are recommended to be cited.
3) For the last sentence of the 2nd paragraph pf the Introduction section, to support the application of ESI in LC-MS and LC-MS/MS analysis, Prof. Li’s work in pharmacokinetics study (10.1002/bmc.5488), and Prof. Wang’s work in DNA metabolism study (10.15252/embj.2023113684) both using ESI-based LC-MS/MS technology are recommended to be cited.
4) In the 3. Conclusions and perspectives section, the sentence, “Today’s Analytical chemistry is not only chemistry”, may need to be revised as “Today’s Analytical Chemistry is not only chemistry”.
5) For some abbreviations of Tables 2, the author is recommended to give the full names.
Author Response
See PDF

Reviewer 3 Report
Comments and Suggestions for Authors
The authors used existing literature from PubMed to summarize the applications of GC-MS and LC-MS/MS in clinical studies. While the manuscript outlines the basic workflows of these mass spectrometry techniques and provides an overview of their usage across various fields, several aspects require further attention for clarity and depth.
1. The title of the paper appears to overstate the scope of the content. The manuscript primarily offers a summary of the frequency of use of these technologies rather than an in-depth exploration of their application in clinical studies. A more accurately descriptive title is recommended.
2. The authors states that the studies utilizing GC-MS and LC-MS do not significantly overlap. However, the rationale for this observation is not adequately articulated. It would be beneficial to discuss the distinct mechanisms of GC-MS and LC-MS, highlighting how the differing chemical properties of analytes influence the choice of technique.
3. To enhance the manuscript's impact, the authors should include specific examples of how GC-MS and LC-MS/MS are applied in clinical research. These examples should illustrate the significance of these techniques in driving scientific discoveries and underscore their unique contributions to the field.
4. A dedicated section comparing GC-MS and LC-MS would help readers understand the reasons behind the selection of one technique over the other in various studies. This section should elaborate on the fundamental differences in methodology and applicability.
5. The comparison of AEA measurements between GC-MS/MS and LC-MS/MS presented in Figure 3 highlights the importance of cross-validation. However, the example chosen may be biased, as many analytes suitable for LC-MS/MS cannot be analyzed using GC-MS/MS. The authors should clarify that the efficacy of cross-validation using GC-MS is highly contingent upon the chemical properties of the analytes in question.
Author Response
See PDF

Round 2
Reviewer 1 Report
Comments and Suggestions for Authors
Dear Author,
the manuscript is still lacking important information about other applications of mass spectrometry besides LC-MS, LC-MS/MS, GC-MS, GC-MS/MS.
In particular, few lines (on regulatory aspects and legislation) are dedicated to high resolution-mass spectrometry applications, which is the real future perspective of mass spectrometry. Based on the only provided search terms, you can not consider this manuscript as a perspective article on mass spectrometry.
Did you consider the role of ICP-MS for the metal analysis in clinical medicine? The only considered hyphenated techniques were GC and LC-MS (or MS/MS), but there are many other techniques involving mass spectrometry.
The article does not provide any future perspectives for mass spectrometry in analytical chemistry, which should be the focus of a "Perspective" article.
The "Methods" section does not report how the literature search was conducted; search terms, inclusion criteria, exclusion criteria were reported.
Author Response
I thank the reviewer for evaluating the revised version.
I have revised the paper in accordance with the suggestions.
All changes are indicated in red.
1) The title was specified by including: GC-MS, LC-MS and ICP-MS.
2) New
2) The review article "Facets of ICP-MS and their potential in the medical sciences-Part 1: fundamentals, stand-alone and hyphenated techniques.
Clases D, Gonzalez de Vega R. Anal Bioanal Chem. 2022 Oct;414(25):7337-7361. doi: 10.1007/s00216-022-04259-1. was considered and cited.
3) Table 1 now included ICP-MS related data from PubMed.
4) The new section 3.6. ICP-MS in clinical science was added.
5) ICP-MS relating issues were addressed in all parts of the paper.
6) The new ICP-MS-related References 34 to 40 were added.
Reviewer 3 Report
Comments and Suggestions for Authors
The questions have been addressed and the manuscript has been sufficiently improved.
Author Response
I thank the reviewer.